# Hydrothermal Synthesis of Nanocrystalline ZrO_2_-8Y_2_O_3_-xLn_2_O_3_ Powders (Ln = La, Gd, Nd, Sm): Crystalline Structure, Thermal and Dielectric Properties

**DOI:** 10.3390/ma14237432

**Published:** 2021-12-03

**Authors:** Radu-Robert Piticescu, Anca Elena Slobozeanu, Sorina Nicoleta Valsan, Cristina Florentina Ciobota, Andreea-Nicoleta Ghita, Adrian Mihail Motoc, Stefania Chiriac, Mythili Prakasam

**Affiliations:** 1National R&D Institute for Non-Ferrous and Rare Metals, INCDMNR-IMNR, 102 Biruintei Blvd., 077145 Pantelimon, Romania; svalsan@imnr.ro (S.N.V.); crusti@imnr.ro (C.F.C.); andreea.lupu@imnr.ro (A.-N.G.); amotoc@imnr.ro (A.M.M.); schiriac@imnr.ro (S.C.); 2CNRS, Université de Bordeaux, ICMCB, UMR 5026, 87 Avenue du Docteur Schweitzer, CEDEX, 33608 Pessac, France; mythili.prakasam@u-bordeaux.fr

**Keywords:** rare-earths doped zirconia, hydrothermal synthesis, crystalline structure, thermal conductivity, impedance spectroscopy

## Abstract

Zirconium dioxide (ZrO_2_) is one of the ceramic materials with high potential in many areas of modern technologies. ZrO_2_ doped with 8 wt.% (~4.5 mol%) Y_2_O_3_ is a commercial powder used for obtaining stabilized zirconia materials (8 wt.% YSZ) with high temperature resistance and good ionic conductivity. During recent years it was reported the co-doping with multiple rare earth elements has a significant influence on the thermal, mechanical and ionic conductivity of zirconia, due complex grain size segregation and enhanced oxygen vacancies mobility. Different methods have been proposed to synthesize these materials. Here, we present the hydrothermal synthesis of 8 wt.% (~4.5 mol%) YSZ co-doped with 4, 6 and 8 wt.% La_2_O_3_, Nd_2_O_3_, Sm_2_O_3_ and Gd_2_O_3_ respectively. The crystalline phases formed during their thermal treatment in a large temperature range were analyzed by X-ray diffraction. The evolution of phase composition vs. thermal treatment temperatures shows as a major trend the formation at temperatures >1000 °C of a cubic solid solutions enriched in the rare earth oxide used for co-doping as major phase. The first results on the thermal conductivities and impedance measurements on sintered pellets obtained from powders co-doped with 8 wt.% Y and 6% Ln (Ln = La, Nd, Sm and Gd) and the corresponding activation energies are presented and discussed. The lowest thermal conductivity was obtained for La co-doped 8 wt.% YSZ while the lowest activation energy for ionic conduction for Gd co-doped 8 wt.% YSZ materials.

## 1. Introduction

Zirconium dioxide (ZrO_2_), also known as zirconia and zirconium oxide, is a ceramic material of great technological importance due to its advanced multidimensional properties, such as: high mechanical strength, chemical inertness, high temperature stability, low thermal conductivity, corrosion resistance, high ionic conductivity and biocompatibility [1,2]. These properties lead to different applications in areas such as gas sensors [3], solid-oxide fuel-cell [4], catalysis [5], thermal barrier coating [6], medical prosthesis [7] and memory devices [8].

Zirconia is found in several polymorphic forms depending on the temperature conditions: monoclinic (m), tetragonal (t) and cubic (c) phases. At temperatures below 1100 °C, the monoclinic phase of pure ZrO_2_ is the stable form. From 1100 °C up to 2370 °C, pure ZrO_2_ has a tetragonal structure, while at temperatures from 2370 °C up to 2706 °C (the melting point) it exists as cubic phase. On cooling, the cubic-tetragonal and tetragonal-monoclinic transformations are accompanied by both a decrease in theoretical density and an expansion of the volume between 3–5%. Therefore, due to the high stresses during the cooling period, pure zirconia undergoes the formation of micro-cracks, strongly affecting its mechanical properties. To prevent these inconveniences, several oxides, such as MgO, CaO and Ln_2_O_3_ (Ln: all transition metals from the lanthanum series in the periodic table of elements) are added to the zirconium oxide to stabilize the tetragonal and/or cubic phases when cooled to sintering temperature. The doped materials thus obtained are generally called partially stabilized zirconia (t or c + t phases are formed) or stabilized zirconia (only c phase formed) [9,10,11,12]. The doping elements do not affect only the structural and mechanical properties but also opto-electronic and phonon behavior due to the compensation of differences between the valence of doping elements and Zr^+4^ matrix with oxygen vacancies according to the rule of charge neutrality [13].

It was shown that using rare earth oxides (REOs) as dopants may avoid grain size coarsening due to interface segregation in doped zirconia ceramic, enhancing its thermo-mechanical properties. It has been theoretically proposed that the grain boundary energy may be correlated with dopant concentration is close to zero, enhancing stability of nanomaterials. The absolute grain boundary energy of Gd-doped nanocrystalline ZrO_2_ vs. grain size can reach a quasi-zero energy state (0.05 J/m^2^), when a critical dopant enrichment is achieved, until a temperature threshold where the dopant re-dissolves in the crystalline bulk [14].

The oxygen vacancies formed due to RE ions dopants were reported to improve the ionic conductivity in a co-doped sintered ((Y_0.75_La_0.25_)_1−x_Gd_x_)_0.18_Zr_0.82_O_2−δ_ system and the energy values for the bulk and grain boundary conductions are strongly related to the ionic radii of the trivalent REs ions and the vacancy concentrations at lower temperatures [15]. Fine-grained ceramics based on solid solutions ZrO_2_-xLn_2_O_3_ (where Ln = Sm and Yb) were obtained by the colloidal chemical synthesis of the powders followed by spark plasma sintering. The addition of lanthanides stabilized the structure and no phase transformation occurred during heating, ensuring a high density of the ceramic obtained [16].

The use of ZrO_2_ doped with mixed REOs is considered as a challenge in designing of SOFCs membranes with high ionic conductivity, long-term temperature performance stability, density and long-term reliability (high strength and high durability). It was reported that co-doping ZrO_2_ with Y_2_O_3_ and CeO_2_ led to high temperature stabilization of tetragonal ZrO_2_ with 6–9 nm crystallite sizes [17]. (ZrO_2_)_0.85_(REO_1.5_)_0.15_ (RE = Y, Sc) solid solutions in pure cubic fluorite structure obtained by a citrate gel method have been proposed as materials with promising properties for SOFCs applications [18].

In the present paper we investigate the thermal conductivities and dielectric properties of ternary ceramic materials from the system ZrO_2_-8Y_2_O_3_-xLn_2_O_3_ powders (Ln = La, Gd, Nd and Sm, x = 4, 6 and 8 wt.%) obtained by the hydrothermal method as a first step in developing new materials with controlled thermal and ionic conductivities.

Different physical, chemical and combined routes were used for nanopowders’ synthesis. The chemical reactions for nanomaterials’ synthesis may take place in a solid, liquid or gaseous state. The main advantage of solid-state reactions is the easy control of the composition, but high activation temperatures are required for grain boundaries’ diffusion which may lead to excessive grain growth. In liquid or gas phase synthesis reactions, the activation energy is many orders of magnitude lower and the synthesis of nanostructured materials can be achieved at lower temperatures avoiding powders coarsening. However, the control of the final chemical composition is more challenging.

Obtaining zirconia nanoparticles has been achieved by hydrothermal [19], spray pyrolysis [20], co-precipitation method [21], sol-gel [22], microwave irradiation [23], combustion [24], ball milling [25], pyrolysis [26], spray pyrolysis [27] and hydrolysis [28], each method developed with the aim to fulfil specific applications.

Hydrothermal process can be defined as a heterogeneous and homogeneous chemical reaction in the presence of a solvent (aqueous or non-aqueous) above room temperature and at pressures higher than 1 atm. in a closed system. These main characteristics lead also to a process with a low environmental impact. Under a hydrothermal reaction, the reactants, in the presence of a mineralizer or solvent, go into solution as complexes due to chemical transport reactions. The hydrothermal process is versatile and takes place in a single step with a minimum energy consumption. Nanosized materials can be obtained in situ, without involving post-synthesis treatment. The presence of surfactants plays an important role in the preparation of highly dispersed, oriented and self-assembled particles of complex or multicomponent materials [29]. The value of the isoelectric point and pH influences the inorganic surface modification in the presence of a surfactant.

The in-situ deprotonation of water molecules from the coordination sphere of metallic ions due to high temperatures and pressures in hydrothermal solutions produces homogeneous nucleation sites for the crystallization of the new phases [30]. The particles’ growth is further controlled by adsorption/desorption and chemical reactions at the interfaces that are topo-chemical processes strongly depending on the interface/surface characteristics. A remarkable advantage of the hydrothermal synthesis is the increased solubility product of compounds in water at elevated temperatures and pressures and the enhanced chemical reactivity of usually insoluble reagents [29]. ZrO_2_ materials doped with different rare earth oxides (REOs) have been recently studied as candidates for new thermal barrier coatings [31,32] for different industrial applications [33,34,35]. At the time they were identified, in the eighteenth and nineteenth centuries, they were considered relatively rare compared to other elements. The term “rare” is actually somewhat misleading; for example, they are more common than lead, copper, gold or silver. REEs are relatively common in the Earth’s crust, but the fact that they are mixed with many other minerals in different concentrations in the soil makes extraction difficult to achieve [36,37,38]. Usually, because they have similar ionic radii and a trivalent charge, REEs are found together in the Earth’s crust, with the exception of Europium, which can occur in a state of valence Eu^2+^ or Cerium as Ce^4+^ [39]. The two largest sources of REE combined in mineral deposits are monazite and bastnaesite. Monazite is a phosphate mineral while bastnaesite is a fluorinated carbonate mineral containing different REEs [40]. The world’s reserves of rare earths are about 120 million metric tons, of which 36.67% are located in China. Significant reserves are also held by the United States, Australia, Brazil, India and Malaysia [41]. China has won a monopoly not only on reserves but also on rare earth production. This has led to a global supply risk since 2010 [42]. Therefore, the European Commission has identified them as critical materials with a significant supply risk, as they play an important role in a number of modern technologies [43]. Different methods were proposed to obtain mixed REOs-doped ZrO_2_ powders.

Mekala and co-workers used the coprecipitation method to synthesize Ce-doped ZrO_2_ and Dy doped ZrO_2_. The nanoparticles thus synthesized showed the tetragonal (t-ZrO_2_) and monoclinic (m-ZrO_2_) phase and the decrease of the crystal size by Dy and Ce doping. Doping with RE elements had different effects on both size (undoped ZrO_2_ had a nanoparticle size of about 12 nm, Dy doping led to a particle size of 9 nm and ZrO_2_ doped with Ce had the smallest particle size at 6 nm) and morphology [44].

The co-precipitation method was also used by Lei Guo et al. to synthesize 1 mol% RE_2_O_3_ and 1 mol% Yb_2_O_3_ co-doped 3.5 mol% Y_2_O_3_ stabilized ZrO_2_ (1RE1Yb-YSZ, RE = La, Nd, Gd and Yb) powders, alternative TBC material. The study showed that Re_2_O_3_ ((RE = La, Nd, Gd and Yb) and Yb_2_O_3_ co-doping has an obvious influence on the thermal conductivity and phase stability of YSZ ceramics for TBC applications. YSZ doping with lower ionic RE has beneficial effects on phase stability, but has a limited effect on reducing thermal conductivity. Therefore, the study reported that among the compounds investigated, 1Gd1Yb-YSZ showed better phase stability and lower thermal conductivity [45]. ZGYbY: ZrO_2_-9.5Y_2_O_3_-5.6Yb_2_O_3_-5.2Gd_2_O_3_ powder, as a new material for TBC applications, was synthesized by a chemical co-precipitation method to increase phase stability of YSZ at higher temperatures, above 1200 °C. The result of the heat treatment at 1300 °C for 50 h indicated ZGYbY powder exhibited excellent stability due to the complete retention of the t ’zirconia tetragonal phase and the transition from tetragonal to monoclinic phase upon cooling while YSZ powder decomposed into two new phases, including cubic and monoclinic zirconia. These results make ZGYbY a promising material for TBC applications in new generations of turbines [46].

Chao Chen et al. used the chemical coprecipitation method and the calcination method to obtain the powders xSc_2_O_3_-1.5Y_2_O_3_-ZrO_2_ (x = 4.5, 5.5 and 6.5, in mol%) and 4.5 mol% Y_2_O_3_-ZrO_2_ in order to test the samples for hot corrosion in molten salts Na_2_SO_4_ + V_2_O_5_ (50/50 %gr.). An excellent hot corrosion resistance was obtained for those ScYSZ ceramics with a higher Sc_2_O_3_ content, such as 6.5Sc1.5YSZ [47].

ZrO_2_YO_1.5_ -TaO_2.5_ powders were prepared by Pitek et al. by inverse co-precipitation of mixed solutions. The zirconium-based material stabilized with 16.6% YO_1.5_ + 16.6% TaO_2.5_ is tetragonal and has phase stability up to at least 1500 °C [48].

Leila Sun et al. obtained, by a co-precipitation technique, ZrO_2_ doped with 7.5 mol% Sc_2_O_3_ and Gd_2_O_3_ (ScGdSZ). Samples co-doped with Gd^3+^·3.7Sc_2_O_3_ and 3.7 Gd_2_O_3_ co-doped ZrO_2_ (in mol%) had the lowest thermal conductivity, which was 20% lower than 7.5 ScSZ and 40% lower than 4, 5 YSZ, respectively [49].

Solution combustion method using Glycine as a fuel, was used by H. C. Madhusudhana et al. to synthesize Gd doped ZrO_2_ (1–9 mol%). In the case of the 7% mol sample of Gd doped ZrO_2_, the results of the impedance spectroscopy show good dielectric properties with a very high dielectric constant (ε′ = 345) and also a high alternative conductivity (σ_ac_ = 0.06837 S·cm^−1^) at 10 MHz, suitable for solid oxide fuel cell applications [1].

Using the solid-state reaction method, X. Song and co-workers obtained ZrO_2_-Ta_2_O_5_-Y_2_O_3_-Ln_2_O_3_ (Ln = Nd, Sm or Gd) powders to investigate the effects of rare earth oxides on thermal structure and properties. There have been significant improvements in thermal conductivity by the addition of rare earth oxides Ln_2_O_3_, but also a decrease in grain growth and specific thermal capacity. Gd_2_O_3_ showed the best efficiency for thermal conductivity [50].

The solid-state reaction method was used by Wang and co-workers to obtain dense x mol% ZrO_2_-Gd_3_NbO_7_ (x = 0, 3, 6, 9, 12). It was observed that the thermal conductivity of ZrO_2_ -Gd_3_NbO_7_ is much lower, at 1.21–1.82 (W·m^−1^·K^−1^) from 25 to 900 °C, compared to La_2_Zr_2_O_7_ (1.50–2.00 W·m^−1^·K^−1^), Y_2_SiO_5_, 8YSZ (2.50–3.00 W·m^−1^·K^−1^), ZrO_2_-DyTaO_4_, ZrO_2_-Y_2_O_3_-Ta_2_O_5_ and Yb_2_O_3_ [51].

The introduction of Sm_2_O_3_ and Yb_2_O_3_ into zirconia results in a higher shrinkage intensity and a lower sintering activation energy. This energy is determined by the influence of Sm^3+^ and Yb^3+^ ions on the diffusion properties of the granule boundaries and by the ratio between the atomic radii of zirconium and lanthanide [16].

Hydrothermal synthesis was one of the methods successfully used for obtaining ZrO_2_ doped with different REOs.

Nanoparticles of zirconia doped with erbium Er^3+^ and ytterbium Yb^3+^ have been synthesized by Jasso and co-workers [52,53] using the hydrothermal route. The XRD analysis for the synthesized samples revealed the face cantered cubic phase of the particles with nanometric crystal size ~ 10 nm.

Powder of zirconia doped with terbium Tb^3+^ were synthesized by Ramos-Guerra et al. [54] by hydrothermal route using different types of stabilizing agents in order to evaluate their influence on the structural and luminescent properties.

The hydrothermal synthesis method was also used by Gionco and co-workers [55] to manufacture nanostructured rare-earth (Ce and Er) doped zirconia nanoparticles in order to evaluate their potential as solar light photocatalysts.

Recently, in order to understand the crystallization mechanism of 5% yttrium stabilized tetragonal zirconia, Song et al. [56] used the hydrothermal technique, starting from precursors with different zirconium coordination.

8YSZ doped with different concentrations of Eu^3+^ was synthesized by the hydrothermal method. It was concluded based on the combination of crystal structure and photoluminescence spectroscopy that 3 mol% Eu^3+^ ions in 8YSZ is the optimal concentration [57].

Ce and Er doped ZrO_2_ (from 0.5 to 10 mol% on an oxide basis) were successfully synthesized by the hydrothermal method [58] in order to investigate the photocatalytic activity caused by the sun. By introducing rare earth ions, the tetragonal (Ce and Er) and cubic (Er, 10 mol%) phases were stabilized.

Nanoparticles with polymorphic phases (m-ZrO_2_, t-ZrO_2_ and c-ZrO_2_) from the ZrO_2_-Me_2_O_3_ system (where Me = Y, Eu, Tb, Sm and Er) with crystallite size in the range 5–25 nm in the form of (spheres, hollow spheres, rods and star structures) with luminescence properties were obtained by the hydrothermal route [59].

## 2. Materials and Methods

### 2.1. Hydrothermal Synthesis

This paper aims to obtain zirconium-based powders doped with 8 wt.% (~4.5 mol.%) Y_2_O_3_ (further abbreviated 8 wt.% YSZ) and co-doped with controlled amounts of La, Nd, Sm and Gd and study the crystalline phase modification after thermal treatment in the temperature range 400–1400 °C, thermal conductivities and ionic conductivities to further assess their potential applications in high temperature thermal barrier coatings (TBC). The 8 wt.% Y_2_O_3_-ZrO_2_ composition was selected based on their commercial use as feed powder in obtaining TBCs in aeronautics or gas turbines [60]. Thus, different REO dopants (4/6/8 wt.% La, 4/6/8 wt.% Nd, 4/6/8 wt.% Sm, 4/6/8 wt.% Gd) were added to the ZrO_2_ doped with 8% Y_2_O_3_, obtaining a number of 12 initial powders further denominated MxZY 4/6/8La, MxZY 4/6/8Nd, MxZY 4/6/8% Sm and MxZY 4/6/8% Gd respectively.

All nanostructured powders were synthesized in a one-step process by the hydrothermal method at moderate temperatures (max. 250 °C) and pressures (max. 40 atm). The temperature was selected based on the estimation of the reaction enthalpy (ΔH), reaction entropy (ΔS), Gibbs free energy (ΔG) and equilibrium constants (log K) for the hydrothermal reactions:(1)$$\begin{array}{cc} 0.984\mathrm{Zr}\left(+4a\right)\;+ & 0.08Y\left(+3a\right)\;+\;0.08\mathrm{Ln}\left(+3a\right)\;+\;4.416\mathrm{OH}\left(-a\right)\\ & =0.984{\mathrm{ZrO}}_{2}+0.04Y_{2}O_{3}+0.04{\mathrm{Ln}}_{2}O_{3}+2.208H_{2}O \end{array}$$
where: Ln-La, Nd, Sm and Gd, (a) denominates the ionic species in aqueous solutions and the stoichiometric coefficients correspond to the highest 8 wt.% Ln as dopants. The estimation has been done using the HSC Chemistry v.10 software and database (Outotec, Pori, Finland). The results are presented in Figure 1a–d. The values obtained for the enthalpies, entropies and Gibbs free energies in Figure 1a–c are very close for the four lanthanides considered in this thermodynamic study, with a difference < 1 kcal.

The synthesis was made from high purity raw materials: Y_2_O_3_ > 99%—Merck, Darmstadt, Germany, La_2_O_3_ ≥ 99.9%—Roth, Karlsruhe, Germany, Nd_2_O_3_ ≥ 99.9%—Alfa Aesar, Kandel, Germany, Sm_2_O_3_ ≥ 99.9%—Alfa Aesar, Gd_2_O_3_ ≥ 99.9%—Alfa Aesar. Kandel, Germany, Zirconium tetrachloride (ZrCl_4_ 99% Merck) was used as raw material for preparing an aqueous Zr (IV) stock solution with programmed Zr concentration. The dissolution of REOs precursors in ZrCl_4_ solution was done under vigorous mechanical stirring until a homogenous clear solution was obtained. Ammonia solution (NH_3_ 25% p.a., Chimreactiv S.R.L.) agent was added as mineralizing until an alkaline suspension with pH ~ 9 was obtained. The suspension thus obtained was transferred to a Teflon vessel of a sealed hydrothermal autoclave reactor (5 L capacity, Berghoff Products + Instruments GmbH, Berghoff, Germany) and subjected to the hydrothermal process, followed by controlled water cooling with the help of a coil inserted in the reaction vessel. A solid precipitate formed, was filtered and washed three times with distilled water to remove soluble impurities and was dried in an oven at 100 °C to constant weight.

A working scheme for an understanding of the entire research activity that has been carried out in this paper is presented in Figure 2.

After the hydrothermal synthesis process, for the assessment of phase stability, the powders MxZY 4/6/8% La, MxZY 4/6/8% Nd, MxZY 4/6/8% Sm and MxZY 4/6/8% Gd, were subjected to calcination at various temperatures (400 °C/800 °C/1000 °C/1200 °C/1400 °C, respectively) for 60 min and their phase compositions were analyzed.

### 2.2. Characterization Methods

REO-co-doped ZrO_2_ powders were chemically analyzed by inductively coupled plasma optical emission spectrometry (Agilent 725 ICP-OES, Agilent Technologies Inc., Colorado Springs, CO, USA) according to ASTM E 1479-99 (2011).

The microstructure of REO-doped ZrO_2_ powders was examined using a BRUKER D8 ADVANCE X-ray diffractometer (Bruker AXS GmbH, Karlsruhe, Germany) with monochromatic Kα radiation, using the Bragg-Brentano diffraction method. Scans were obtained in the range 2θ 4–74° with a step size of 0.02° every 2.5 or 6 s. To identify the phases contained in the samples, the data processing was performed using the software package DIFFRAC.SUITE.EVA launched 2016 by Bruker AXS Company, Karlsruhe, Germany; SLEVE + 2020 and the ICDD PDF-4 + 2020 database published by the International Center for Diffraction Data (ICDD).

The morphology of the powder samples was investigated by scanning electron microscopy (SEM) using a high-resolution microscope Quanta 250 (FEI Company, Eindhoven, The Netherlands), incorporated with Energy Dispersive X-Ray Spectrometer, produced by EDAX (Mahwah, NJ, USA), consisting of ELEMENT Silicon Drift Detector Fixed, Element EDS Analysis Software Suite APEX™ 1.0, EDAX, Mahwah, NJ, USA.

The room-temperature thermal conductivity was measured using the hot disk method (TPS 2200 Hot Disk, Hot Disk AB, Göteborg, Sweden) on pairs of cylindrical pellets (approximate 20 mm diameter and 5 mm height) thermal treated at 1200 °C. A 2 mm diameter Kapton sensor (code 7577) sandwiched between the two cylindrical replicate samples was used to generate heat and monitor the temperature evolution of the sample. The thermal conductivity was calculated using the data processing software developed by the manufacturer.

Impedance spectroscopy measurements were performed on Probostat equipment (NORECS, Oslo, Norway) placed into a vertical tubular furnace (ELITE, Leicestershire, UK). As impedance analyzer an Array M3500A instrument (Array, Nanjing, China) was used in order to measure in AC mode. Impedance measurements were conducted in a frequency interval between 1 kHz and 100 kHz. Temperature range was established between 200 °C and 900 °C at a 100 °C interval. For analysis and impedance data fitting, NOVA 2.1 software from Methrom Autolab. (Metrohm Autolab B.V., Utrecht, The Netherlands) was used.

Differential scanning calorimetry coupled with thermal gravimetry analysis (DSC-TG) has been performed on a SETARAM SETSYS Evolution equipment (SETARAM Instrumentation, Caluire, France) in inert atmosphere. Samples were introduced in alumina crucibles and heated with a heating rate of 10 K/min up to 1450 °C, then cooled to room temperature with the same rate. The experimental data were processed with the help of Calisto software v.1.097 (SETARAM Instrumentation, Caluire, France).

## 3. Results

### 3.1. Chemical Analysis

The chemical analysis of the powders synthesized under hydrothermal conditions is in accordance with the designed compositions. The concentration of Zr, Y and Ln in mother liquor resulting from the filtering of the hydrothermal reaction products and in the wash-waters was in all cases <10^−3^ g·L^−1^.

### 3.2. XRD Analysis

The XRD spectra of initial and thermal treated samples are presented in Figure 3a–d.

The qualitative phase analysis is presented in Table 1.

The crystallite sizes of the initial hydrothermally synthesized powders have been calculated from the broadening of [111] peak of the cubic phase using the Scherrer formula:(2)$$d=\frac{K\lambda }{\beta \cos\theta }$$
where *d* is the mean size of the crystalline domains; *K* is a dimensionless shape factor, with the typical value of about 0.9; *λ* is the X-ray wavelength; *β* is the line broadening at half the maximum intensity (FWHM), after subtracting the instrumental line broadening, in radians; and *θ* is the Bragg angle. The results are presented in Table 2.

All initial hydrothermal powders consist of cubic Yttrium Zirconium Oxide as the major phase and monoclinic M-ZrO_2_ (Baddeleyite) as the secondary phase, with the exception of 8 wt.% YSZ co-doped with Nd that consists only of the cubic phase.

All samples thermally treated at 400 °C and 800 °C have a similar phase composition. The major phase is a solid substitution solution with the cubic structure of ZrO_2_ (C-(ZrO_2_)_ss_) in which Zr^+4^ is isomorphically substituted by Ln^+3^ in similar proportions.

With increasing thermal treatment temperatures, a secondary solid solution with tetragonal symmetry (T-(ZrO_2_)_SS_) specific to intermediate temperatures is formed. At 1200 °C and 1400 °C the formation of pyrochlore structures (Pyr-RE_2_Zr_2_O_7_ where RE—Y and Ln) is observed. However, for samples MxZY-Nd with 4, 6 and 8 wt. Nd at 1200 °C and 1400 °C, MxZY-Sm with 6 and 8 wt.% Sm at 1000 °C, 1200 °C and 1400 °C and MxZY-Gd with 8 wt.% Gd only the cubic solid solution (ZrO_2_)_SS_-(Ln_0.14_Y_0.14_Zr_0.72_)O_1.86_ phase was observed.

### 3.3. SEM Analysis

Scanning electron microscopy images show that nanopowders with irregular shapes are formed in all systems and no significant grain growth is observed after calcination. Figure 4 presents, as an example, the morphology of 8 wt.% YSZ powders co-doped with 6% Ln_2_O_3_ as obtained by hydrothermal process and after calcination to 1000 °C. The grain sizes of all samples are summarized in Table 3.

### 3.4. DSC Analysis

Thermal analysis was used to analyze the thermal stability of powder and phase transformations during thermal treatment of hydrothermally synthesized powders. As an example, the TG and DSC graph of the MxZy4%Gd powder heated from ambient temperature to 1450 °C is shown in Figure 5, with the specification that the graphs of the other powders are similar. The mass losses for each individual system revealed in the TG curve are shown in Table 4.

### 3.5. Thermal Conductivity

The thermal conductivity at room temperatures for ZrO_2_ co-doped with 8 wt.% Y and 6 wt.% Ln (Ln = La, Sm, Nd, Sm and Gd) are presented in Figure 6. The values obtained are in the range of 0.305 W/mK for La co-doped pellets to 0.38 W/mK for Gd co-doped pellets. The values are similar with those measured by Shi et al. for 4 wt.% RE_2_O_3_ (RE = La, Yb, Ce and Gd) and Y_2_O_3_ co-doped ZrO_2_ powder synthesized by a sol–gel method [61]. Thermal conductivities of RE-YSZ (RE = La, Yb, Ce and Gd) tetragonal powders were 0.5181, 0.4215, 0.4851, 0.5187 and 0.5347 W/mK, respectively. Other thermal properties are given in Table 5.

### 3.6. Impedance Spectroscopy

Figure 7a–d presents the Nyquist plots for samples ZrO_2_ co-doped with 8 wt.% Y_2_O_3_ and 6 wt.% Ln_2_O_3_ (Ln = La, Sm, Nd, Sm, Gd) measured at 300, 400, 500 and 600 °C respectively. It may be observed that at temperatures below 500 °C only one semi-circle is presented. In the spectra registered at 500 °C a second semi-circle emerged and the formation of two semi-circles is clearly seen in all spectra from 600 °C. The two semi-circular arcs observed at 600 °C indicate the grain and grain boundary contribution respectively [62]. The values obtained for the conductivity of samples analysed were further used to calculate the activation energy of the conduction according to the equation:(3)$$\sigma T=\sigma_{0}\exp\;\left(-\mathrm{Ea}/\mathrm{kT}\right)$$
where E_a_ is the activation energy for electrical conduction, σ is electrical conductivity, σ_0_ pre-exponential factor, k is Boltzmann constant and T is temperature, from the Arrhenius plot presented in Figure 8. Activation energy calculated from the slopes of the least-squares fit of lnσ vs 1000/T in the temperature range 300–600 °C (573–873 K) are presented in Table 6.

## 4. Discussions

The hydrothermal method provides an efficient route to obtain ZrO_2_ nanopowders with controlled chemical composition by co-doping with Y, La, Nd, Sm and Gd. SEM analysis shows the formation of ceramic powders with irregular shape and grain sizes in the range 1.3–70 µm with no significant grain growth during thermal treatment at temperatures up to 1000 °C.

All initial hydrothermally synthesised powders consist of a mixture of Cubic Yttrium Zirconium Oxide as the major phase and Monoclinic ZrO_2_ with the typical structure of natural mineral Baddeleyite, excepting initial powders doped with 4, 6 and 8 wt.% Nd_2_O_3_ where only the cubic phase was formed. These results confirm previous results [63], showing that the hydrothermal synthesis of yttria-doped zirconia nanopowders favours the stabilization of the cubic phase at low temperatures, induced by the uniform distribution of temperature and pressure during the nucleation process.

After thermal treatment at 400 °C, the phase composition is homogenized and all samples consist of the mixture of cubic and monoclinic phases previously mentioned. DSC-TG analysis confirm that around 400 °C the hydroxyl groups from the precipitated powders are removed and a fully crystalised powder is formed. With further increase the thermal treatment temperature to 1000 °C, a tetragonal ZrO_2_ solid solution is formed as a new phase in all powders, excepting 8 wt.% YSZ co-doped with 6 and 8 wt.% Sm_2_O_3_.

At 1200 °C and 1400 °C the formation of a new cubic solid solutions and pyrochlore structures (Pyr-RE_2_Zr_2_O_7_ where RE—Y and Ln) is observed. The phase compositions obtained at high thermal treatment temperatures are in good agreement with those predicted from the study of phase diagrams briefly summarised in [64].

In the case of 8 wt.% YSZ co-doped with 4, 6 and 8 %wt. Nd_2_O_3_, 8YSZ co-doped with 6 and 8 wt.% Sm_2_O_3_ and 8 wt.% YSZ co-doped with 8 wt.% Gd_2_O_3_, a cubic solid solution was observed as single phase. The evolution of crystalline phases vs. composition and temperature shows a clear trend toward the formation of cubic solid solutions with different compositions depending. The higher stability of cubic zirconia with increasing temperature is well known and is explained by the lower surface energy of the cubic symmetry. The cubic phases formed in the 8 wt.% YSZ-co-doped with 4, 6 and 8 wt.% Ln_2_O_3_ (Ln = La, Nd, Sm and Gd) can be classified into three types of solid solutions, depending on the dopant concentration: (Ln_0.07_Y_0.14_Zr_0.79_)O_1.90_; (Ln_0.11_Y_0.14_Zr_0.75_)O_1.88_ and (Ln_0.14_Y_0.14_Z_r0.72_)O_1.86_.

The increase in dopant concentration leads to decreasing the height and widening of diffraction maxima. These results could be explained by the following fact: the increase of the dopant concentration favours the introduction of micro-tensions at the level of the elementary cell and the formation of nanometric crystallites. More detailed works and deconvolutions are further needed to evaluate these changes induced in the elementary cell in a multiphase material.

The low thermal conductivity of 8 wt.% YSZ co-doped with 4, 6 and 8 wt.% Ln_2_O_3_ may be related to the complex phase composition and the formation of cubic pyrochlore that was reported as low thermal conductive materials [65]. However, the results obtained by the hot disk method should be further checked with other standardized methods, such as flash DSC.

Impedance spectrometry indicates that at about 600 °C the migration of the charge carriers take place through different mechanisms, due to the modification of the grain boundaries and bulk contributions vs. dopant used, leading to the decrease of the total electrical resistance of samples with increasing temperature. The dopants modify the grain resistance and grain boundary behaviour within the studied frequency range. This is in accordance with previous studies showing that in the case of doping with Gd the conductivity presents a rapid increase with temperature [66]. The formation of complex cubic solutions in the case of co-doping 8 wt.% YSZ with LnO_3_ is enhanced by the ratio between Ln^3+^ ionic radii and Y^3+^ ionic radius leading to modification of the intergranular energy and grain boundaries diffusion properties [16].

With increasing temperature, the grain boundary contribution becomes the conduction limiting step. The activation energies have values of the same order of magnitude for all 8 wt.% YSZ-6 wt.% Ln_2_O_3_ samples studied, with a lower value of 1.09 eV for 6 wt.% Gd co-doped materials. Compared to the experimental results presented in [67] where activation energies for bulk conductivity of YSZ doped with 1 and 2 wt.% Yb and Gd were in the range 0.68–0.84 eV, it may be considered that the higher concentration of co-doping elements, the higher association of vacancies concentrations with lattice defects, influencing the total conductivities of materials. These results will be further used to understand the structure modification induced by co-doping ZrO_2_ with various Rare Earth Oxides.

The influence of the sintering procedure (classic uniaxial pressing followed by air sintering and spark plasma sintering) is a work in progress to validate the experimental results regarding thermal conductivities and optimise their processing to assess their potential applications as materials for high temperature TBCs.

## 5. Conclusions

Zirconia powders co-doped with 8 wt.% Y_2_O_3_ and 4/6/8 wt.% Ln_2_O_3_ (Ln = La, Nd, Sm and Gd) were obtained by hydrothermal synthesis in one-step process at moderate temperatures (max. 250 °C) and pressures (max. 40 atm). Nanocrystalline powders consisting of cubic Yttrium Zirconium Oxide as a major phase and monoclinic M-ZrO_2_ (Baddeleyite) as a secondary phase were obtained, excepting YSZ doped with Nd that consist only of a cubic phase. The phase evolution of powders during thermal treatment in the range 400–1400 °C shows the progressive formation of different solid solution by isomorphically substitution of Zr^4+^ with Ln^3+^. A cubic ZrO_2_ solid solution is the major phase and secondary phases with generic formula (Ln_0.07_Y_0.14_Zr_0.79_)O_1.90_; (Ln_0.11_Y_0.14_Zr_0.75_)O_1.88_ and (Ln_0.14_Y_0.14_Z_r0.72_)O_1.86_ are formed depending on the dopant concentration.

At temperatures 1200 °C and 1400 °C, the formation of cubic solid solutions with Pyrochlore structures (Pyr-RE_2_Zr_2_O_7_ where RE—Y and Ln) is observed for all compositions, with the exception of samples co-doped with 4, 6 and 8 wt.% Nd, 6 and 8 wt.% Sm and 8 wt.% Gd, when only the cubic (ZrO_2_)_SS_-(Ln_0.14_Y_0.14_Zr_0.72_)O_1.86_ phase was observed.

Low values of the thermal conductivities were measured by the hot plate method at room temperature. The impedance spectroscopy allowed calculation of the activation energy of conduction for zirconia powders co-doped with 8 wt.% Y_2_O_3_ and 6 wt.% Ln_2_O_3_, with the lower value 1.09 eV for Gd co-doped materials.

Further works are in progress to correlate the thermal conductivities and activation energies with the complex structures induced by each co-doping element and assess the potential applications in TBCs.

## Figures and Tables

**Figure 1 materials-14-07432-f001:**
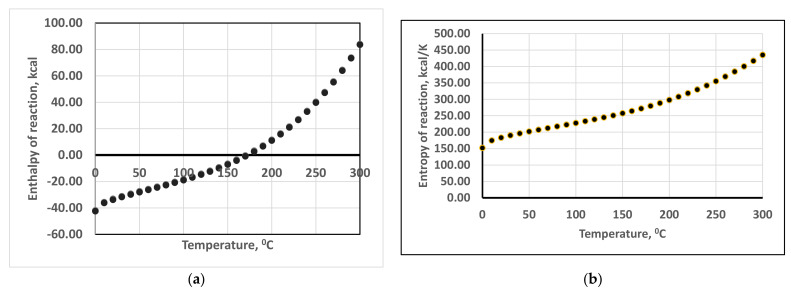
(**a**) Enthalpies of reactions (1), (**b**) entropies of reactions (1), (**c**) Gibbs free energy of reactions (1), (**d**) equilibrium constants of reactions (1). Series 1 corresponds to La^3+^, series 2 to Nd^3+^, series 3 to Sm^3+^ and series 4 to Gd^3+^.

**Figure 2 materials-14-07432-f002:**
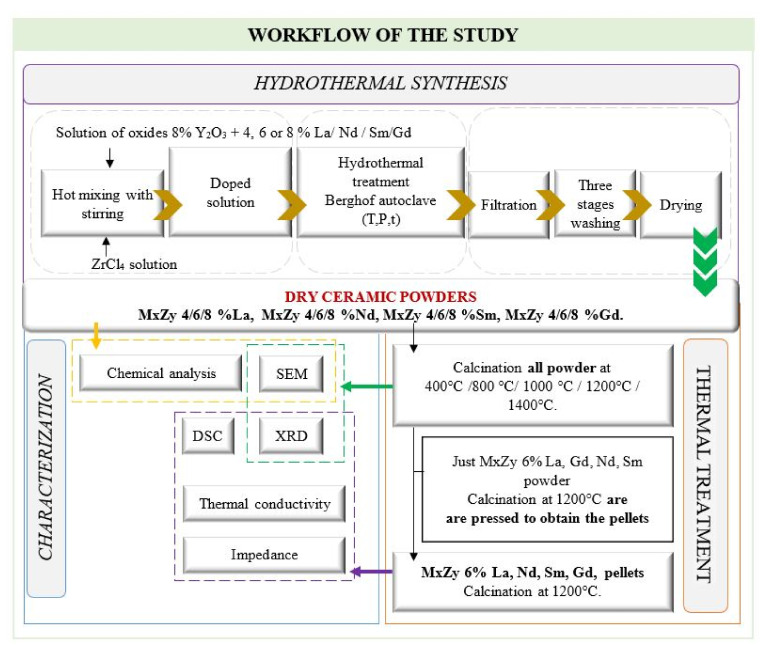
Schematic workflow of the study.

**Figure 3 materials-14-07432-f003:**
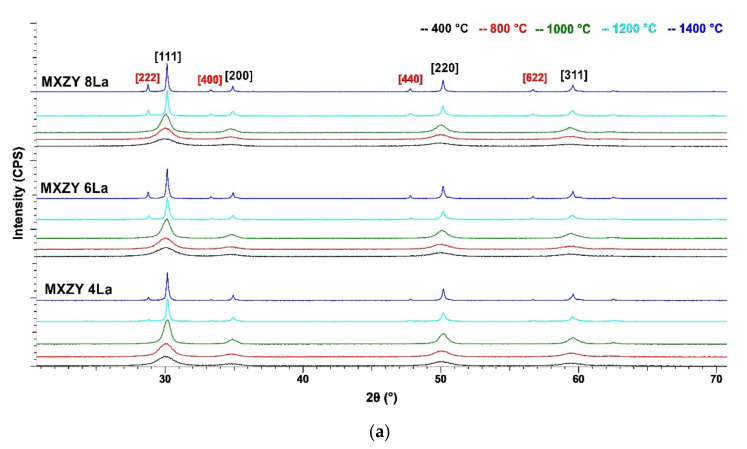
(**a**) XRD spectra of 8 wt.% YSZ powders co-doped with 4, 6 and 8 wt.% La_2_O_3_ thermal treated at different temperatures; (**b**) XRD spectra of 8 wt.% YSZ powders co-doped with 4, 6 and 8 wt.% Nd_2_O_3_ thermal treated at different temperatures; (**c**) XRD spectra of 8 wt.% YSZ powders co-doped with 4, 6 and 8 wt.% Sm_2_O_3_ thermal treated at different temperatures; (**d**) XRD spectra of 8 wt.% YSZ powders co-doped with 4, 6 and 8 wt.% Gd_2_O_3_ thermal treated at different temperatures.

**Figure 4 materials-14-07432-f004:**
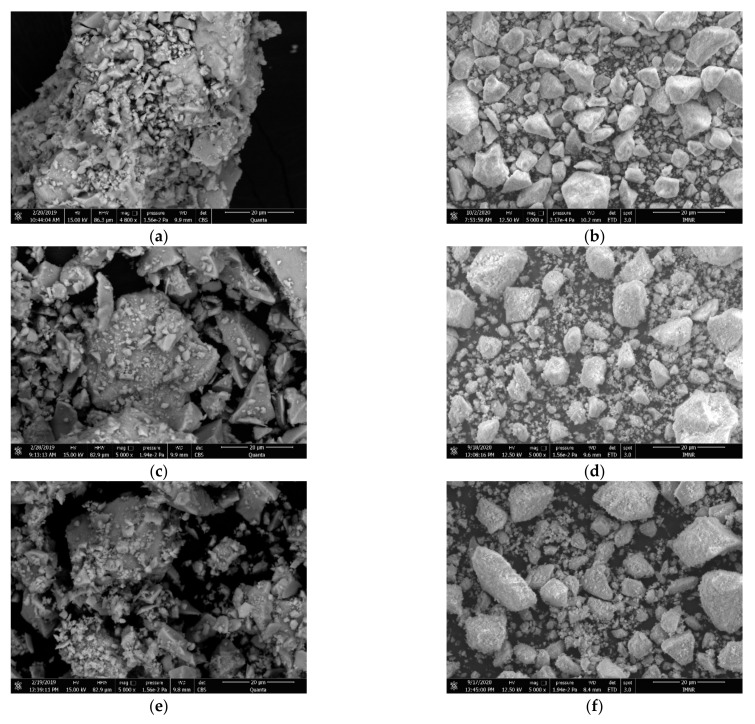
SEM images for 6% MxZYLn powders as obtained and calcined at 1000 °C. (**a**) MxZY-6La initial powders; (**b**) MxZY-6La calcined 1000 °C powders; (**c**) MxZY-6Nd initial powders; (**d**) MxZY-6Nd calcined 1000 °C powders; (**e**) MxZY-6Sm initial powders; (**f**) MxZY-6Sm calcined 1000 °C powders; (**g**) MxZY-6Gd initial powders; and (**h**) MxZY-6Gd calcined 1000 °C powders.

**Figure 5 materials-14-07432-f005:**
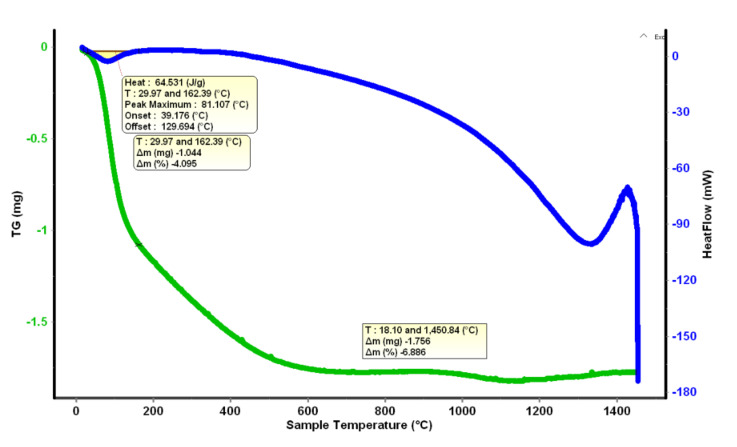
DSC-TG analysis of MxZY4% Gd powder.

**Figure 6 materials-14-07432-f006:**
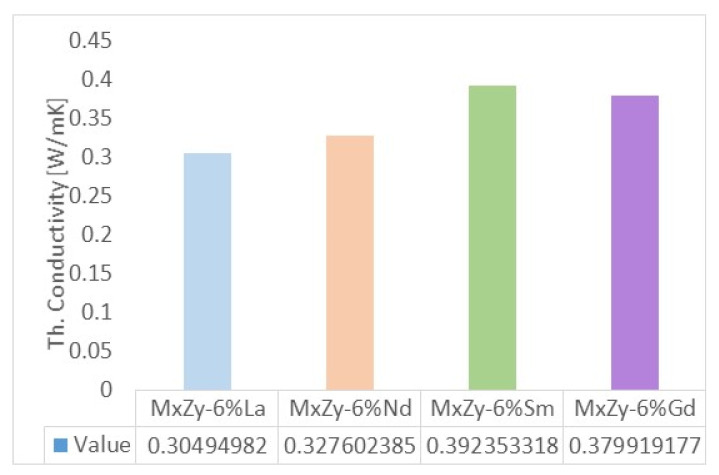
Thermal conductivity for ZrO_2_ co-doped with 8 wt.% Y-6% Ln (Ln = La, Nd, Sm, Gd).

**Figure 7 materials-14-07432-f007:**
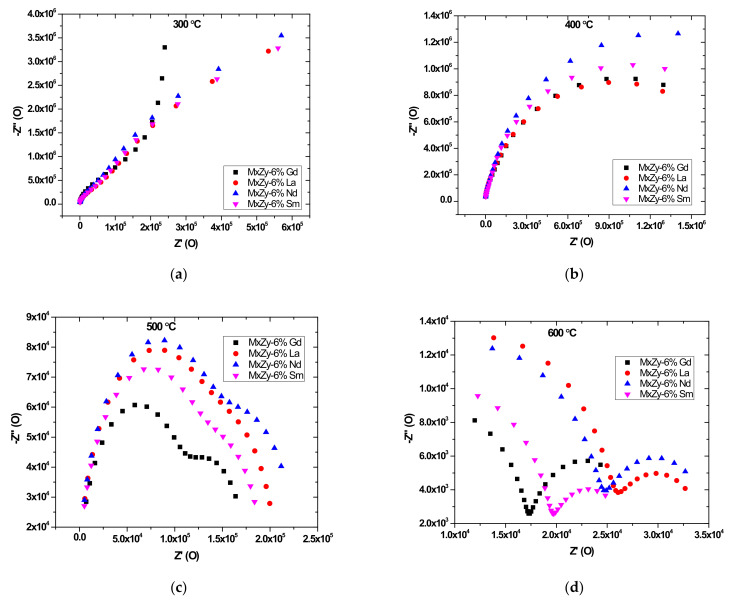
(**a**) Nyquist plots for samples ZrO_2_ co-doped with 8 wt.% Y_2_O_3_ and 6 wt.% Ln_2_O_3_ at 300 °C; (**b**) Nyquist plots for samples ZrO_2_ co-doped with 8 wt.% Y_2_O_3_ and 6 wt.% Ln_2_O_3_ at 400 °C; (**c**) Nyquist plots for samples ZrO_2_ co-doped with 8 wt.% Y_2_O_3_ and 6 wt.% Ln_2_O_3_ at 500 °C; and (**d**) Nyquist plots for samples ZrO_2_ co-doped with 8 wt.% Y_2_O_3_ and 6 wt.% Ln_2_O_3_ at 600 °C.

**Figure 8 materials-14-07432-f008:**
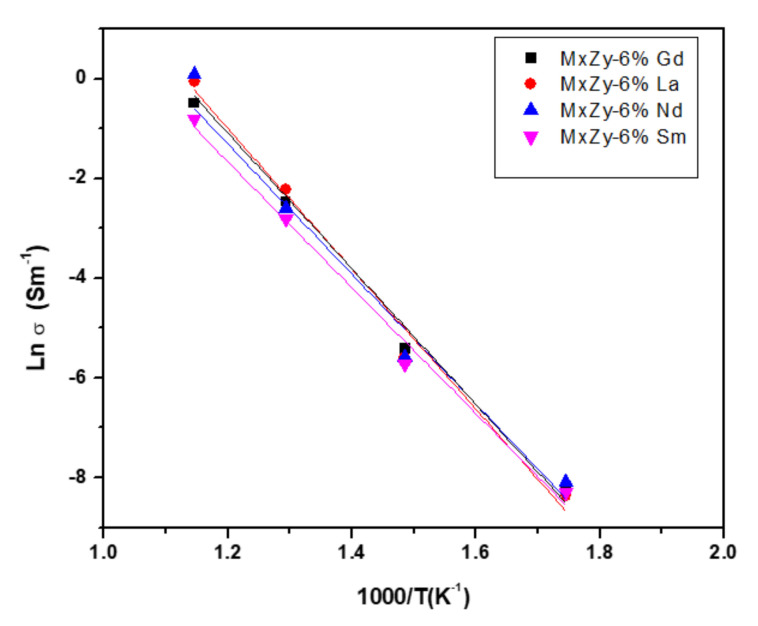
Arrhenius plots for ZrO_2_ co-doped with 8 wt.% Y_2_O_3_ and 6 wt.% Ln_2_O_3_ at 600 °C.

**Table 1 materials-14-07432-t001:** Qualitative phase analysis of ZrO_2_ co-doped with 8 wt.% Y_2_O_3_ and 4/6/8 wt.% Ln_2_O_3_ powders initials and after calcination.

Sample	% Dopant	Phase Composition					
		Initial	400 °C	800 °C	1000 °C	1200 °C	1400 °C
MxZY-La	4	C M	C-(ZrO_2_)_SS 1_ M-(ZrO_2_)_SS_	C-(ZrO_2_)_SS 1_ M-(ZrO_2_)_SS_	C-(ZrO_2_)_SS 1_ M-(ZrO_2_)_SS_ T-(ZrO_2_)_SS_	C-(ZrO_2_)_SS 1_ M-(ZrO_2_)_SS_ T-(ZrO_2_)_SS_ Pyr	C-(ZrO_2_)_SS 1_ M-(ZrO_2_)_SS_ T-(ZrO_2_)_SS_ Pyr
	6	C M	C-(ZrO_2_)_SS 2_ M-(ZrO_2_)_SS_	C-(ZrO_2_)_SS 2_ M-(ZrO_2_)_SS_	C-(ZrO_2_)_SS 2_ M-(ZrO_2_)_SS_ T-(ZrO_2_)_SS_	C-(ZrO_2_)_SS 2_ M-(ZrO_2_)_SS_ T-(ZrO_2_)_SS_ Pyr	C-(ZrO_2_)_SS 2_ M-(ZrO_2_)_SS_ T-(ZrO_2_)_SS_ Pyr
	8	C M	C-(ZrO_2_)_SS 3_ M-(ZrO_2_)_SS_	C-(ZrO_2_)_SS 3_ M-(ZrO_2_)_SS_	C-(ZrO_2_)_SS 3_ M-(ZrO_2_)_SS_ T-(ZrO_2_)_SS_	C-(ZrO_2_)_SS 3_ M-(ZrO_2_)_SS_ T-(ZrO_2_)_SS_ Pyr	C-(ZrO_2_)_SS 3_ M-(ZrO_2_)_SS_ T-(ZrO_2_)_SS_ Pyr
MxZY-Nd	4	C	C-(ZrO_2_)_Ss 4_ M-(ZrO_2_)_SS_	C-(ZrO_2_)_SS 4_ M-(ZrO_2_)_SS_	C-(ZrO_2_)_SS 4_ M-(ZrO_2_)_SS_	C-(ZrO_2_)_SS 4_	C-(ZrO_2_)_SS 4_
	6	C	C-(ZrO_2_)_SS 5_ M-(ZrO_2_)_SS_	C-(ZrO_2_)_SS 5_ M-(ZrO_2_)_SS_	C-(ZrO_2_)_SS 5_ M-(ZrO_2_)_SS_ T-(ZrO_2_)_SS_	C-(ZrO_2_)_SS 5_	C-(ZrO_2_)_SS 5_
	8	C	C-(ZrO_2_)_SS 6_ M-(ZrO_2_)_SS_	C-(ZrO_2_)_SS 6_ M-(ZrO_2_)_SS_	C-(ZrO_2_)_SS 6_ M-(ZrO_2_)_SS_ T-(ZrO_2_)_SS_	C-(ZrO_2_)_SS 6_	C-(ZrO_2_)_SS 6_
MxZY-Sm	4	C M	C-(ZrO_2_)_SS 7_ M-(ZrO_2_)_SS_	C-(ZrO_2_)_SS 7_ M-(ZrO_2_)_SS_	C-(ZrO_2_)_SS 7_ M-(ZrO_2_)_SS_ T-(ZrO_2_)_SS_	C-(ZrO_2_)_SS 7_ M-(ZrO_2_)_SS_ T-(ZrO_2_)_SS_	C- (ZrO_2_)_SS 7_ M-(ZrO_2_)_SS_ T-(ZrO_2_)_SS_
	6	C M	C-(ZrO_2_)_SS 8_ M-(ZrO_2_)_SS_	C-(ZrO_2_)_SS 8_ M(ZrO_2_)_SS_	C-(ZrO_2_)_SS 8_	C-(ZrO_2_)_SS 8_	C-(ZrO_2_)_SS 8_
	8	C M	C-(ZrO_2_)_SS 9_ M-(ZrO_2_)_SS_	C-(ZrO_2_)_SS 9_ M-(ZrO_2_)_SS_	C-(ZrO_2_)_SS 9_	C-(ZrO_2_)_SS 9_	C-(ZrO_2_)_SS 9_
MxZY-Gd	4	C M	C-(ZrO_2_)_SS 10_ M-(ZrO_2_)_SS_	C-(ZrO_2_)_SS 10_ M-(ZrO_2_)_SS_	C-(ZrO_2_)_SS 10_ M-(ZrO_2_)_SS_ T-(ZrO_2_)_SS_	C-(ZrO_2_)_SS 10_ T-(ZrO_2_)_SS_	C-(ZrO_2_)_SS 10_ T-(ZrO_2_)_SS_
	6	C M	C-(ZrO_2_)_SS 11_ M-(ZrO_2_)_SS_	C-(ZrO_2_)_SS 11_ M-(ZrO_2_)_SS_	C-(ZrO_2_)_SS 11_ M-(ZrO_2_)_SS_ T-(ZrO_2_)_SS_	C-(ZrO_2_)_SS 11_ T-(ZrO_2_)_SS_	C-(ZrO_2_)_SS 11_ T-(ZrO_2_)_SS_
	8	C M	C-(ZrO_2_)_SS 12_ M-(ZrO_2_)_SS_	C-(ZrO_2_)_SS 12_ M-(ZrO_2_)_SS_	C-(ZrO_2_)_SS 12_ M-(ZrO_2_)_SS_ T-(ZrO_2_)_SS_	C-(ZrO_2_)_SS 12_	C-(ZrO_2_)_SS 12_

Abbreviations used: C—Cubic Yttrium Zirconiu Oxides, PDF 01-077-2286; M—Monoclinic ZrO_2_ Baddeleyite, PDF 00-036-0420; C-(ZrO_2_)_SS 1_—Cubic (_La0.07_Y_0.14_Zr_0.79_)O_1.9_, PDF 04-011-8534; M – (ZrO_2_)_SS_ —Monoclinic ZrO_2_ Baddeleyite, PDF 00-036-0420; T-(ZrO_2_)_SS_ —Tetragonal, PDF 01-078-3348; Pyr-Cubic RE_2_Zr_2_O_7–_La_2_Zr_2_O_7,_ PDF 04-017-6784; C-(ZrO_2_)_SS 2_—Cubic (La_0.11_Y_0.14_Zr_0.75_)O_1.88,_ PDF 04-011-8534; C-(ZrO_2_)_SS 3_—Cubic (La_0.14_Y_0.14_Zr_0.72_)O_1.86_, PDF 04-011-8534; C-(ZrO_2_)_SS 4_—Cubic (Nd_0.07_Y_0.14_Zr_0.79_)O_1.9,_ PDF 04-011-8534; C-(ZrO_2_)_SS 5_—Cubic (Nd_0.11_Y_0.14_Zr_0.75_)O_1.88_, PDF 04-011-8534;.C-(ZrO_2_)_SS 6_—Cubic (Nd_0.14_Y_0.14_Zr_0.72_)O_1.86,_ PDF 04-011-8534; C-(ZrO_2_)_SS 7_—Cubic (Sm_0.07_Y_0.14_Zr_0.79_)O_1.9 86_, PDF 04-011-8534; C-(ZrO_2_)_SS 8_—Cubic (Sm_0.11_Y_0.14_Zr_0.75_)O_1.88 86,_ PDF 04-011-8534; C-(ZrO_2_)_SS 9_—Cubic (Sm_0.14_Y_0.14_Zr_0.72_)O_1.86_, PDF 04-011-8534; C-(ZrO_2_)_SS 10_—Cubic (Gd_0.07_Y_0.14_Zr_0.79_)O_1.9,_ PDF 04-011-8534; C-(ZrO_2_)_SS 11_—Cubic (Gd_0.11_ Y_0.14_Zr_0.75_)O_1.88_, PDF 04-011-8534; C-(ZrO_2_)_SS 12_—Cubic (Gd_0.14_Y_0.14_Zr_0.72_)O_1.86,_ PDF 04-011-8534.

**Table 2 materials-14-07432-t002:** Mean crystallite sizes of the initial hydrothermally synthesized ZrO_2_ doped powders.

Sample	% Dopant	*d* (nm)	Sample	% Dopant	*d* (nm)
MxZY-La	4	6	MxZY-Sm	4	5.4
	6	7		6	5.6
	8	7		8	5.1
MxZY-Nd	4	6.3	MxZY-Gd	4	5.7
	6	6		6	5.7
	8	5.7		8	5.6

**Table 3 materials-14-07432-t003:** Grain sizes of 4, 6, 8% MxZyLn powders, initial and calcinated at 1000 °C.

Sample		Concentration of Dopant (%)		
		4	6	8
MxZYLa	Initial	2.20–42.81 nm	2.14–14.53 nm	1.91–24.37 nm
	Calcined 1000 °C	1.29–17.55 nm	1.68–9.72 nm	1.51–20.25 nm
MxZYNd	Initial	2.35–68.34 nm	2.02–52.31 nm	1.83–14.54 nm
	Calcined 1000 °C	1.45–14.60 nm	1.91–12.91 nm	1.45–14.60 nm
MxZYSm	Initial	2.65–11.49 nm	2.05–30.48 nm	1.62–18.73 nm
	Calcined 1000 °C	1.36–17.71 nm	1.76–22.06 nm	1.55–13.83 nm
MxZYGd	Initial	2.71–58.39 nm	1.78–52.27 nm	12.41–19.08 nm
	Calcined 1000 °C	1.26–16.96 nm	1.35–20.69 nm	1.52–23.3 nm

**Table 4 materials-14-07432-t004:** Mass losses (wt.%) for MxZy%Ln powders.

Sample	H_2_O Loss	Total Loss	Sample	H_2_O Loss	Total Loss
MxZY4% La	5.142	8.46	MxZy4% Sm	4.876	8.243
MxZY6% La	5.164	9.406	MxZy6% Sm	4.733	7.979
MxZY8% La	6.68	11.42	MxZy8% Sm	5.924	9.985
MxZY4Nd	4.859	8.168	MxZy4% Gd	4.095	8.224
MxZY6Nd	4.693	8.132	MxZy6% Gd	4.994	7.987
MxZY8Nd	6.299	10.233	MxZy8% Gd	5.295	8.352

**Table 5 materials-14-07432-t005:** Thermal properties of ZrO_2_ co-doped with 8 wt.% Y-6% Ln (Ln = La, Nd, Sm, Gd).

Sample	Th. Conductivity (W/m·K)	Th. Diffusivity (mm^2^/s)	Volumetric Spec. Heat (MJ/m^3^·K)	Spec. Heat (MJ/m^3^·K)
MxZY-6%La	0.3049 ± 0.0079	0.2603 ± 0.0075	1.1715 ±0.0104	0.4048 ± 0.0036
MxZY-6%Nd	0.3276 ± 0.0004	0.2899 ± 0.0018	1.1297 ± 0.0077	0.3502 ± 0.0023
MxZY-6%Sm	0.3923 ± 0.0019	0.2766 ± 0.0053	1.4185 ± 0.0216	0.3987 ± 0.0060
MxZY-6%Gd	0.3799 ± 0.0012	0.2799 ± 0.0084	1.3577 ± 0.0379	0.3809 ± 0.0106

**Table 6 materials-14-07432-t006:** Activation energy of conduction for ZrO_2_ co-doped with 8 wt.% Y_2_O_3_-6% Ln_2_O_3_ (Ln = La, Nd, Sm and Gd).

Sample	Ea (eV)
MxZY-6%La	1.12
MxZY-6%Nd	1.21
MxZY-6%Sm	1.17
MxZY-6%Gd	1.09

## Data Availability

The data presented in this study are available on request from the corresponding authors.

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
