# Peer review of "Hydrothermal Synthesis of Nanocrystalline ZrO2-8Y2O3-xLn2O3 Powders (Ln = La, Gd, Nd, Sm): Crystalline Structure, Thermal and Dielectric Properties"

_materials, 2021, doi:10.3390/ma14237432_

Round 1

Reviewer 1 Report

The manuscript contains a large and comprehensive review of zirconia ceramics. A detailed analysis of several similar papers, including comments on experiments and results, is provided. Obviously it is not a full review of zirconia, the 5 congresses on Science and Technology of zirconia is not referenced. A careful reading of the manuscript prompts some comments:

Line 50: "cooled from", not "cooled to"

156: "radii", not "rays"

585: "(SEM Quanta...)" is already in the experimental section.

586: ...for doped powder with different amounts of dopant and powder doped with various amounts of dopant... ???

Table 6: missing the unit of mass loss

Figure 6, impedance diagrams: Z' (O), -Z'' (O)? Should be Ohm or Ohm.cm, unless all samples had the same dimensions. 

It is usual the impedance diagram be plotted with the maximum value in the ordinate (Z'')= half the maximum value in the abscissa (Z'); the plot be a rectangle with the Z' axis dimension the double of the Z'' axis dimension. This would allow researchers on impedance spectroscopy to better evaluate the diagrams (e.g. depressed semicircles could infer inhomogeneity). Arrows pointing to some data in the impedance diagrams could help the analysis.

Figure 7, caption should be ...total conductivity.

601: "at 1200..." instead of "at temperatures 1200..."

All over the manuscript there are sometimes commas instead of dots for numbers (e.g. line 314, 316 317, 318, etc.); "Pyrochlore" instead of "pyrochlore"

It os an exhaustive experimental work and a detailed description of some previous work, good for people (mainly students) going to this research area. Some corrections are suggested to improve the readability of the manuscript. 

Author Response

Thank you very much for your effort, observations and suggestions. 

  1. The manuscript contains a large and comprehensive review of zirconia ceramics. A detailed

analysis of several similar papers, including comments on experiments and results, is provided. Obviously it is not a full review of zirconia, the 5 congresses on Science and Technology of zirconia is not referenced.

Reply: Considering the observations from reviewers 2 and 3, we decided to reduce the size of literature review regarding synthesis and include former part 2 in Introduction. We hope this is not affecting the quality of the paper.

  1. A careful reading of the manuscript prompts some comments:

Line 50: "cooled from", not "cooled to"

156: "radii", not "rays"

585: "(SEM Quanta...)" is already in the experimental section.

586: ...for doped powder with different amounts of dopant and powder doped with various amounts of dopant... ???

Table 6: missing the unit of mass loss

601: "at 1200..." instead of "at temperatures 1200..."???

All over the manuscript there are sometimes commas instead of dots for numbers (e.g. line 314, 316 317, 318, etc.); "Pyrochlore" instead of "pyrochlore" ???

Reply: We corrected the errors. After re-numbering table 6 is now table 4.

  1. Figure 6, impedance diagrams: Z' (O), -Z'' (O)? Should be Ohm or Ohm.cm, unless all samples had the same dimensions.  It is usual the impedance diagram be plotted with the maximum value in the ordinate (Z'')= half the maximum value in the abscissa (Z'); the plot be a rectangle with the Z' axis dimension the double of the Z'' axis dimension. This would allow researchers on impedance spectroscopy to better evaluate the diagrams (e.g. depressed semicircles could infer inhomogeneity). Arrows pointing to some data in the impedance diagrams could help the analysis. Figure 7, caption should be ...total conductivity.

Reply: Thank you very much for the discussion. We used pellets with the same diameter and height and the units are Ohm. We used the representation from fig. 7a-d and 8 (after renumbering) as it was proposed in the papers mentioned in different papers such as reference 68, in order to compare the results with other existing data for similar materials.

Reviewer 2 Report

In this manuscript, the authors have reported their work on the hydrothermal synthesis of 8 wt% YSZ co-doped with 4-8 wt% La, Nd, Sm, and Gd. The synthesized materials were characterized for their phase identification, thermal conductivity, activation energy of ionic conduction, etc. Although the authors have presented a considerable amount of work, the manuscript lacks conciseness and satisfactory discussion of the results. Therefore, this manuscript will require a major revision before it could be reconsidered for publication. I have the following specific comments and suggestions for improvements.

  1. The entire work is based on 8 wt% YSZ and the authors claim that it is “a gold standard for obtaining cubic stabilized zirconia materials (8YSZ) with high temperature resistance and good ionic conductivity.” Unfortunately, this is wrong. It is 8 mol% YSZ that shows the highest ionic conductivity among YSZ doped with different amounts of yttria and is widely used for solid oxide fuel cells and other applications. Therefore, this work will not have much significance for the stated applications. The authors should correct their statement and explain why they chose 8 wt% YSZ as the base material.
  2. The abstract should be rewritten to explain major findings and contributions rather than the background of the work.
  3. Section 2 (Synthesis of REOs doped zirconia nanopowders) should be removed. The most essential information in this section could be summarized in a paragraph or two and added in the Introduction section.
  4. Figure 1: Which dopant(s) the curves in a-c represent? What do the Series 1-4 represent in d?  
  5. Table 5: The scale bars should be legible in the SEM images.
  6. Table 5 and Figure 5: Use periods (.) to separate the decimals.
  7. Figure 4 should be redrawn for legibility.
  8. Table 6 (p. 22): What is meant by “dust MxZy%Ln”? What are the units for H2O loss and Total loss?
  9. The numbering of tables should be corrected (Table 6 appears twice).
  10. Most of the results shown in figures and tables lack interpretation and discussion of their significance. The authors should note that merely a compilation of results cannot be a research paper. 

Author Response

Thank you for your comments. Please find attached our reply.

Reviewer 3 Report

The paper entitled ‘Hydrothermal synthesis of nanocrystalline ZrO2-8Y2O3-xLn2O3 powders (Ln = La, Gd, Nd, Sm): crystalline structure, thermal and dieclectric properties’ by R.R. Piticescu et al. presents a study on codoped stabilized zirconia (YSZ) based on a large number of synthesized sample and several characterization techniques. However, the present form of the manuscript does not allow to accept it. The three main reason of the reject are:

1/ The (very long) introduction part is not adapted to the proposed paper. It looks like to a part of a review article, and not to the introduction of a research paper.

2/ The presented results are not sufficiently detailed, or interpreted. As example, the DRX mineralogical results (Table 3) are hardly comprehensible, the Table is indigestible with many indicated phases for some samples, that doesn’t correlate with the presented powder patterns (Figures 3). Some chemical compositions are given for solid solution without indication on the way to obtain them, and they are a confusion between the pyrochlore definite composition described as a solid solution in the conclusion. Another example of not really comprehensible description of the obtained results concerns the impedance spectroscopy: results show a two steps mechanism with two semicircles for samples heat treated at 500°C and 600°C. this is mentioned by the authors, but they are no indication on the way the two circles have been extrapolated, and there are a unique Ea value for these samples. In a general point of view, the treatment of experimental data is never clear.

3/ The discussion part is simply missing. A short part of the discussion can be found in the conclusion, that take into account a minority of the experimental results. Some results are only presented in the Results part, but they are never discussed and compared with others technics: SEM, TGA, thermal conductivity and impedance spectroscopy.

Author Response

Thank you very much for your comments. Please find attached our reply.

Reviewer 4 Report

In general, the work is well conducted from a methodological point of view. The different synthesis methodologies along with various characterization techniques were carried out. The topic is of importance for the specific area of catalysis and I believe that the manuscript has a good chance of attracting the attention of a more general audience. In short, I accept the paper with no revision.

Author Response

Thank you very much for your considerations. 

Round 2

Reviewer 2 Report

The revised manuscript looks in a better shape. The authors have made the manuscript somewhat concise and improved the quality of figures and tables. They have also added a separate Discussion section to discuss and interpret the results. However, their explanation for 8 wt% YSZ is not convincing and rather misleading. Although 8 wt% YSZ (which is ~4.5 mol% YSZ) is commonly used for thermal barrier coatings, it is not a fully stabilized cubic zirconia with high oxygen ion conductivity. The authors should first be clear about this fact and revise their description.

  1. When specifying the YSZ composition, use both wt% and mol% so that there would not be any confusion for the readers. Also, I suggest avoiding the use of acronym 8YSZ because it may mean either 8 wt% YSZ or 8 mol% YSZ depending on the context.
  2. Clarify the significance of 8 wt% YSZ for SOFCs and oxygen ion conducting membranes. Because partially stabilized zirconia is good only as a mechanical support for solid oxide cells, I do not see any relevance of this composition for ionic conductivity studies. For high ionic conductivity, one must use fully stabilized or 8 mol% YSZ (~14 wt% YSZ).

Author Response

Thank you again very much for your valuable observations and suggestions. 

  1. In order to avoid any confusion regarding the composition of co-doped powders we used the abbreviation 8 wt.% YSZ, explaining in the beginning of paragraph 2.1 that This paper aims to obtain zirconium-based powders doped with 8 wt% (~4.5 mol.%) Y2O3 and co-doped with controlled amounts of La, Nd, Sm and Gd (further abbreviated 8 wt.%. YSZ) and study……….
  2. We removed the potential SOFCs application, considering both the actual use in this purpose of 8 mol.% Y2O3 and the lower activation energy obtain as a result of interactions between vacancies and other defects produced by co-doping with more REOs. In the end of discussion part (rows 508-514) we modified the text and added the following text to explain how we will further use impedance spectrometry measurements:

The results (Note: of ionic conductivities) will be further used to understand the structure modification induced by co-doping ZrO2 with various Rare Earth Oxides.

  1. We did also corrections to keep the same sample codes along the whole paper:
  • 4 (Y instead of y);
  • Fig, 6 and 7: ZrO2 co-doped with 8 wt.% Y2O3 and 6 wt.% Ln2O3 instead of 6 wt. % Ln

Reviewer 3 Report

Dear authors, I refuse to reconsider again this paper which I have already refused. Let us trust with others reviewers.

Best regards

Author Response

Dear reviewer,

Thank you for the valuable comments from the first evaluation round.